# Research on Belt Deviation Fault Detection Technology of Belt Conveyors Based on Machine Vision

Xiangfan Wu [1] , Chusen Wang [2], Zuzhi Tian [2,*] , Xiankang Huang [2] and Qian Wang [1]

1 School of Mechanical and Electrical Engineering, Xuzhou University of Technology, Xuzhou 221018, China
2 School of Mechatronic Engineering, China University of Mining and Technology, Xuzhou 221116, China
* Correspondence: tianzuzhi2023@163.com

**Abstract:** Traditional belt deflection detection devices for underground belt conveyors in coal mines have problems, such as their single function, poor fault location and analysis accuracy, low automation level, and low reliability. In order to solve the defects of traditional detection devices, the belt deviation faults of the underground belt conveyor transport process require to be detected effectively and reliably. This paper proposes a belt deviation detection method based on machine vision. This method makes use of a global adaptive high dynamic range imaging method to complete the brightness enhancement processing of the underground image. Then the straight-line features of the conveyor belt edges are extracted using Canny edge detection and the Hough transform algorithm. In addition, a dual-baseline localization judgment method is proposed to realize the identification of band bias faults. Finally, a test bench for belt conveyor deviation was built. Testing experiments for different deviations were conducted. The accuracy of the tape deviation detection reached 99.45%. The method proposed in this study improves the reliability of belt deviation fault detection of underground belt conveyors in coal mines and has wide application prospects in the field of coal mining.

**Keywords:** belt conveyor; belt deviation detection; machine vision; fault identification





## 1. Introduction

China's available coal reserves are rich—the current storage capacity accounts for 13.3% of the world's total, ranking third in the world, second only to the United States and Russia [1,2]. The proportion of production is now 50.8% of the world's total coal production. At present, China is in the stage of rapid development. There is no evidence of reduced demand for coal resources, so it is necessary to strengthen coal production safety [3,4] and to improve the level of mining technology. It is important to progressively achieve automated mining [5,6] to ensure the healthy development of the coal mining industry.

As an important part of coal mine production, the safety and stability of the operation of underground coal mine belt conveyors directly affect the safety and efficiency of coal production [7–9]. Belt conveyors can enable the long-distance, large capacity, continuous and stable transport of raw coal to the outside. This is an important part of the coal mining system. However, belt conveyors work under high load and high intensity for long periods in a harsh environment. They are prone to belt deviation failure during transport, which may cause serious accidents and affect the healthy development of the coal industry [10–12].

Most traditional conveyor belt deviation fault detection devices use triggered structural changes to enable detection. However, they are prone to fail when operating in the harsh environment of the underground for a long time, which leads to a reduction in their reliability. At present, equivalent detection systems have been developed both at home and abroad for the detection of belt deflections of belt conveyors in underground coal mines [13]. However, the problems are that the systems have a single function, poor fault location and analysis accuracy, a low automation level, and low reliability. Moreover, the systems developed so far are generally very costly and are not easy to maintain [14,15].

Therefore, how to effectively and reliably detect and warn of belt deviation faults in the transport process of mining belt conveyors is an urgent problem needing to be solved in the safe production of coal energy.

Liu [16] established an adaptive segmentation model and a belt offset quantification model for continuous online detection of the conveyor belt deviation status. Their results showed that the degree of conveyor belt deviation can be quantitatively calculated and the deviation status can be objectively evaluated. Xu Cheng et al. [17] proposed an improved 8-neighbourhood seed-filling algorithm for detecting the edge position of belt conveyors in complex environments. It can quickly detect belt edge information in complex environments and ensure that the conveyor belt is able to work continuously and efficiently. Wang et al. [18] proposed an improved edge detection algorithm based on a Canny operator and morphological processing, and a belt-positioning algorithm based on Hough line detection. This algorithm solves the problem of difficult-to-extract straight lines from the edge of the belt and adapts to the positioning of the belt under complex operating conditions. Zhang [19] proposed a novel conveyor belt deviation monitoring method based on deep learning. The method is realized by improving the output results of a general target detection network, YOLOv5, such that the network is enhanced with the ability to detect straight lines instead of bounding boxes.

Considering the current status of research, in order to overcome the above problems associated with the belt deflection detection technology of underground belt conveyors in coal mines, this paper proposes a machine-vision-based belt deflection detection method for underground belt conveyors in coal mines. Firstly, a global adaptive high-dynamic-range imaging method is used to enhance the brightness of acquired conveyor photos. Then the straight-line features of the conveyor belt edges are extracted using Canny edge detection and the Hough transform algorithm. Finally, a dual-baseline positioning judgment method is proposed. The method is used to enable the identification of conveyor belt deviation faults. In this study, a belt conveyor deflection test bed was built to detect different deflections to verify the feasibility and accuracy of this method. Based on machine vision, the research innovatively applies Canny edge detection and the Hough transform algorithm to belt conveyor deviation detection. We aim to obtain a stable and efficient belt conveyor deviation detection method to improve the reliability of underground belt conveyor monitoring systems.

## 2. Theory

### 2.1. Image Low-Light Processing

Due to the poor lighting conditions in the underground tunnels of coal mines, the images captured by the camera sometimes have the problem of dark brightness. This problem affects the processing and analysis of the images and reduces the accuracy of fault monitoring. Auxiliary lighting can improve this situation but can only partially solve the problem of poor image brightness. So, it is necessary to use a technical means of low illumination brightening processing of underground images.

In this study, a global adaptive high dynamic range imaging (HDR) [20,21] method is used to brighten coal mine underground belt conveyor images and coal mine roadway images and recover the details of their darker parts, which is represented in Equation (1) [22]:

$$g(x,y) = \frac{\log[f(x,y)/\overline{f} + 1]}{\log(f_{\max}/\overline{f} + 1)} \tag{1}$$

where $f_{\max}$ is the maximum value of the pixel intensity, $f(x,y)$ is the intensity of each pixel, and $\overline{f}$ is the logarithmic mean of the pixel intensity. $\overline{f}$ can be obtained from Equation (2):

$$\overline{f} = \exp\left\{\frac{1}{H \cdot W}\sum \log[\sigma + f(x,y)]\right\} \tag{2}$$

where $\sigma$ denotes the correction value set to prevent the appearance of black dots with pixel intensity 0 in the image, $H$ is the number of pixels in the picture length, and $W$ is the number of pixels in the picture width.

Figure 1 shows images of an underground belt conveyor before and after processing using the global adaptive high dynamic range imaging method.

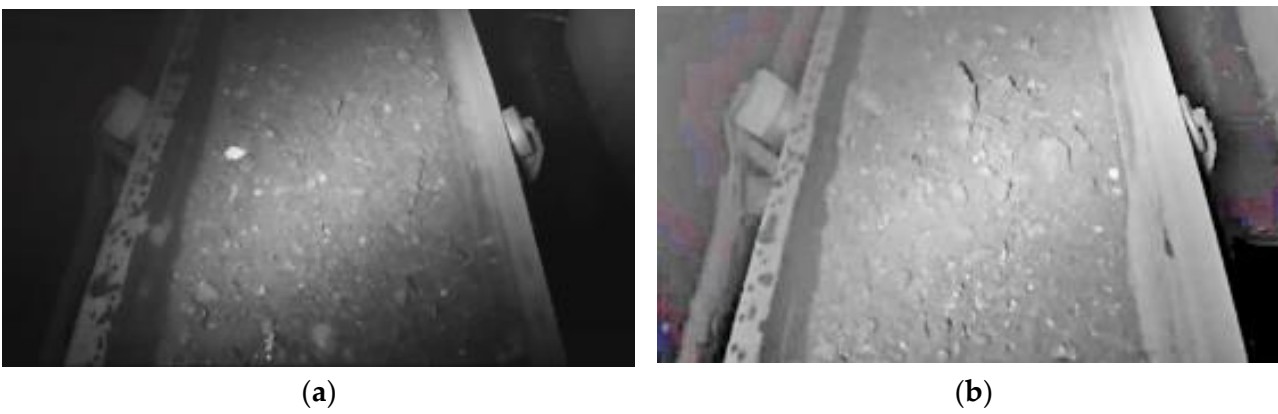

(**a**)  (**b**)

**Figure 1.** Conveyor image before and after low-light processing. (**a**) Before treatment; (**b**) After treatment.

Comparing the effect before and after processing in Figure 1, it can be seen that the brightness of the image affected by the global adaptive high dynamic range imaging method improves. In order to more clearly see the effect of brightening, a grey-scale histogram was extracted from the original image and the brightened image; the results are shown in Figure 2.

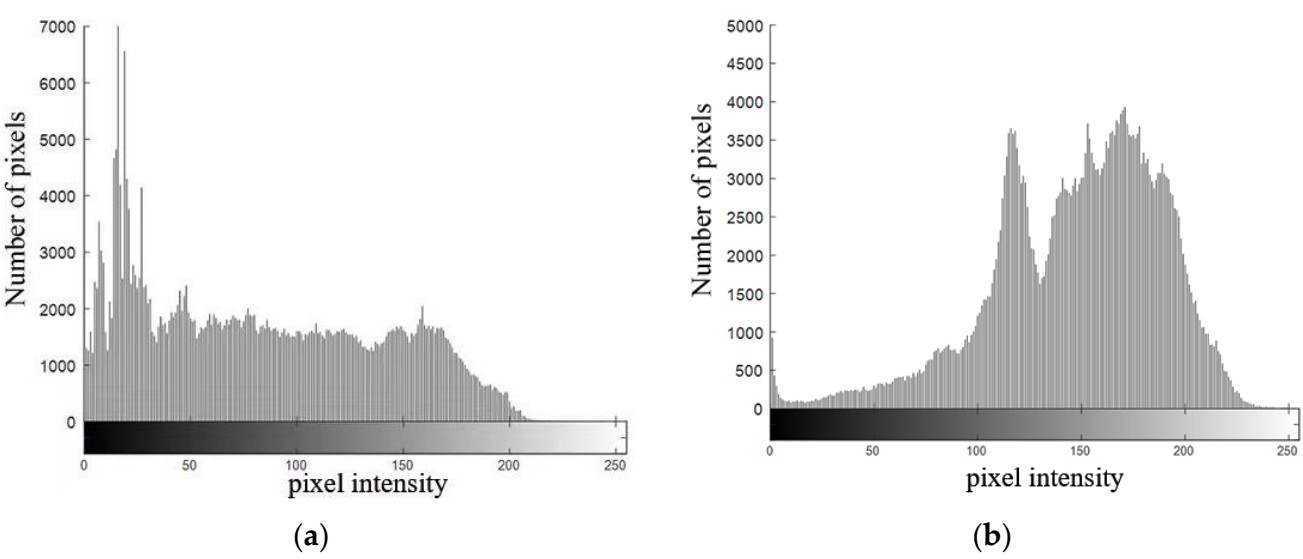

(**a**)  (**b**)

**Figure 2.** Grey-scale histogram of conveyor image before and after low illumination processing. (**a**) Before treatment; (**b**) After treatment.

It can be clearly seen from Figure 2 that the pixel intensity of the image is generally enhanced after the brightening process, the brightness problem of the image is effectively improved, and the clarity of the roadway image is also significantly improved.

### 2.2. Belt Edge Feature Extraction

(1) Canny edge detection

The conveyor belt edge is the most important basis for judging belt deviation faults. It is also the most effective method to judge belt deviation faults by analyzing the position of

the conveyor belt edges. In this study, we chose the Canny edge detection algorithm [23] to extract the features of the conveyor belt bilateral edges. The Canny edge detection algorithm can be divided into the following four steps [24]:

① Gaussian filtering of images

Edge detection algorithms generally work on the basis of the derivative of the intensity of the image pixels. The derivative is more sensitive to the noise of the image. So, it is necessary to use the appropriate filtering means for filtering the image processing, using the traditional Canny edge detection with Gaussian filtering to smooth the image.

② Calculate the gradient value and direction

In calculating the gradient value of an image, the horizontal operator $Soble_x$ and the vertical operator $Soble_y$ are generally calculated using convolution with the input image to obtain the horizontal and vertical components of the gradient $d_x$ and $d_y$ [25], as shown in the following equation:

$$Soble_x = \begin{bmatrix} 1 & 0 & -1 \\ 2 & 0 & -2 \\ 1 & 0 & -1 \end{bmatrix} \tag{3}$$

$$Soble_y = \begin{bmatrix} 1 & 2 & -1 \\ 0 & 0 & 0 \\ 1 & -2 & -1 \end{bmatrix} \tag{4}$$

$$d_x = f(x,y) \cdot Soble_x(x,y) \tag{5}$$

$$d_y = f(x,y) \cdot Soble_y(x,y) \tag{6}$$

The principle of image edge gradient calculation is shown in Figure 3.

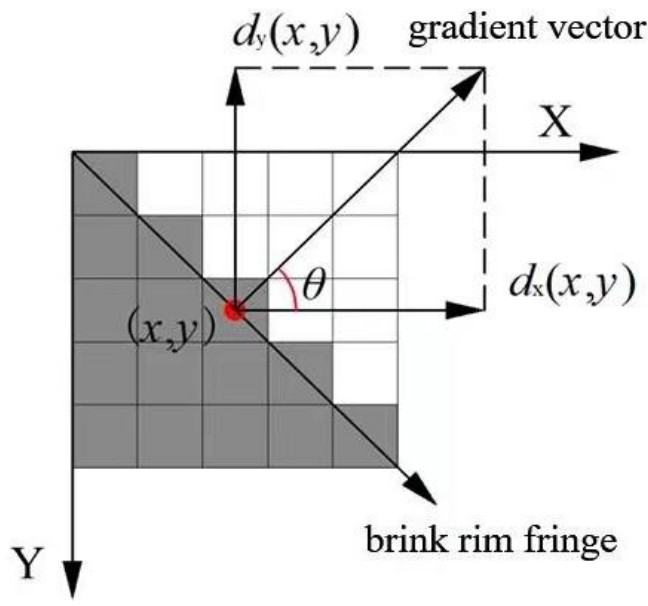

**Figure 3.** Schematic diagram of image edge gradient calculation.

Based on the horizontal and vertical components of the gradient $d_x$ and $d_y$, the gradient magnitude $M(x,y)$ at the $(x,y)$ coordinates can be further obtained as shown in Equation (7):

$$M(x,y) = \sqrt{d_x^2(x,y) + d_y^2(x,y)} \tag{7}$$

The gradient direction $\theta_M$ at the $(x, y)$ coordinates is:

$$\theta_M = \arctan\frac{d_y}{d_x} \tag{8}$$

③ Non-extreme value suppression

The result obtained after gradient calculation does not identify the pixel as the target feature point; only when the maximum value is obtained at the pixel can it be identified as the target feature point. So, the operation of non-maximum value suppression needs to be carried out by comparing the center pixel of the neighborhood with the pixel of the gradient direction. If the pixel is larger than the pixel of the gradient direction, then it is retained. The rest of the pixel gradients are assigned to 0, and vice versa. The pixel gradient is assigned to 0, and the pixel with the largest value of the gradient is retained. Non-extremely large value suppression can retain the pixel with the largest gradient in the neighborhood to effect edge refinement.

When the gradient angle is not located orthogonally or diagonally, then there is practically no pixel point in the gradient direction. The location can be regarded as a sub-pixel point, and the gradient value of the sub-pixel point is obtained by interpolating the values of the pixels on both sides. An approximation algorithm is proposed in the Canny edge detection algorithm. The algorithm can approximate the gradient direction within a certain angular range to a fixed direction, as shown in Figure 4. For example, when the gradient direction is located in the direction $-22.5°\sim22.5°$, the approximation replaces the pixel points with $0°$ direction and so on.

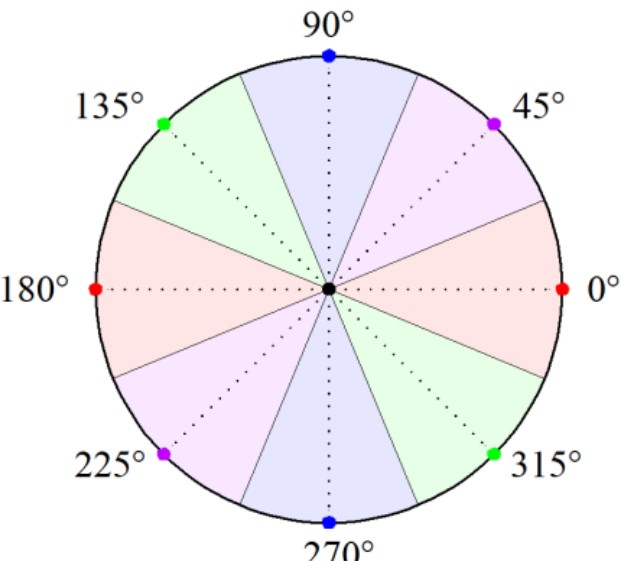

**Figure 4.** Schematic diagram of non-extreme value suppression.

④ Double-threshold filtering and edge connectivity

The edge of the image obtained by non-extremely large value suppression is more accurate. However, because of the noise and image color changes some edge pixels are not removed. So, we need find the means to remove this part of the excess pixel points. To obtain a more accurate edge, the double-threshold filtering method is an effective method that has been proposed to solve this kind of problem. The principle of the method is as follows:

a. To set the high and low thresholds, the Canny edge detection algorithm recommends keeping the ratio of the high and low thresholds between 2 and 3;

b. If the gradient magnitude of a pixel is greater than the high threshold, the pixel is retained as an edge pixel;

c. If the gradient magnitude of a pixel is less than the low threshold, the pixel will be removed;

d. If the gradient magnitude of a pixel is between a low threshold and a high threshold, the pixel is determined to be an edge pixel or not based on the connectivity, and, if the pixel is connected to an identified edge pixel, the pixel is retained as an edge pixel, otherwise it is removed.

(2)  Hough transform linear detection

The Hough transform is an algorithm that detects and displays straight lines in a graph on the basis of completing the edge detection [26]. The image field of view used in this paper is small and the conveyor belt edges can be approximated as straight lines, and straight-line extraction can be carried out effectively using the Hough transform.

The working principle of the Hough transform is to transform the point on the edge line from a right-angled coordinate system to a polar coordinate system. The straight line in the edge line is converted to a point in the polar coordinate system. As shown in Figure 5, the points $P_1$ and $P_2$ in the Cartesian coordinate system are two trigonometric curves in the polar coordinate system after Hough transformation. The two curves intersect at a point $Q$, which can represent the straight line in the Cartesian coordinate system where the points $P_1$ and $P_2$ are located. All the points on the straight line are intersected at the point $Q$ after transforming them to the polar coordinate system, and the points on the straight line satisfy the functional relationship expressed by Equation (9):

$$\rho_i = x_i \cos \theta_i + y_i \sin \theta_i \tag{9}$$

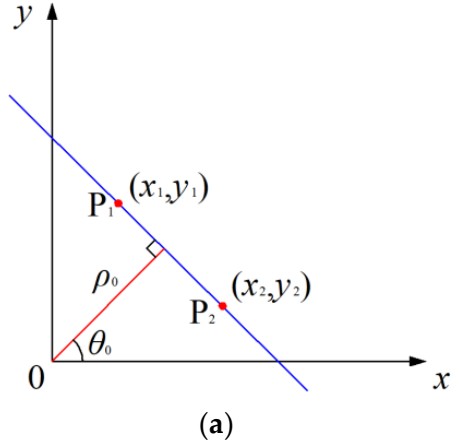

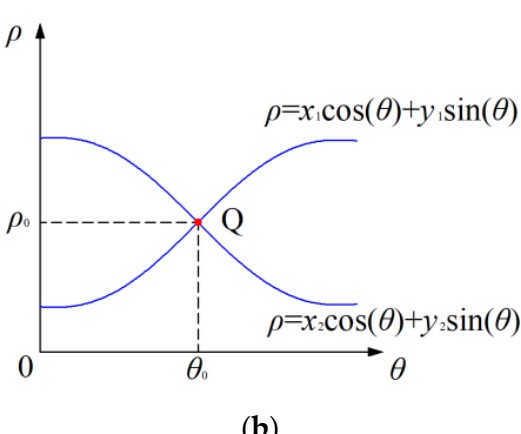

(**a**)  (**b**)

**Figure 5.** Schematic diagram of Hough transform coordinate transformation. (**a**) Right-angle coordinate dotted line; (**b**) Polar coordinate dotted line.

The number of intersecting curves reflects the length of a straight line in a right-angled coordinate system. The Hough transform achieves the objective of straight-line detection using such a transformation.

The Hough transform does not detect and display all straight lines when performing straight-line detection, because a threshold is set artificially when performing the Hough transform. Only when the length of a straight line is greater than or equal to the threshold will the line be judged as a valid straight line and displayed. As shown in Figure 6, the Hough transform follows the flow shown in the diagram when a straight line is detected.

When Hough transform straight line detection is performed on the results of Canny edge detection, the threshold can be adjusted according to the length of the straight line that actually needs to be detected. The detection effect is such that the interfering lines can be removed as much as possible, and only straight lines where the edge of the conveyor belt is located can be retained.

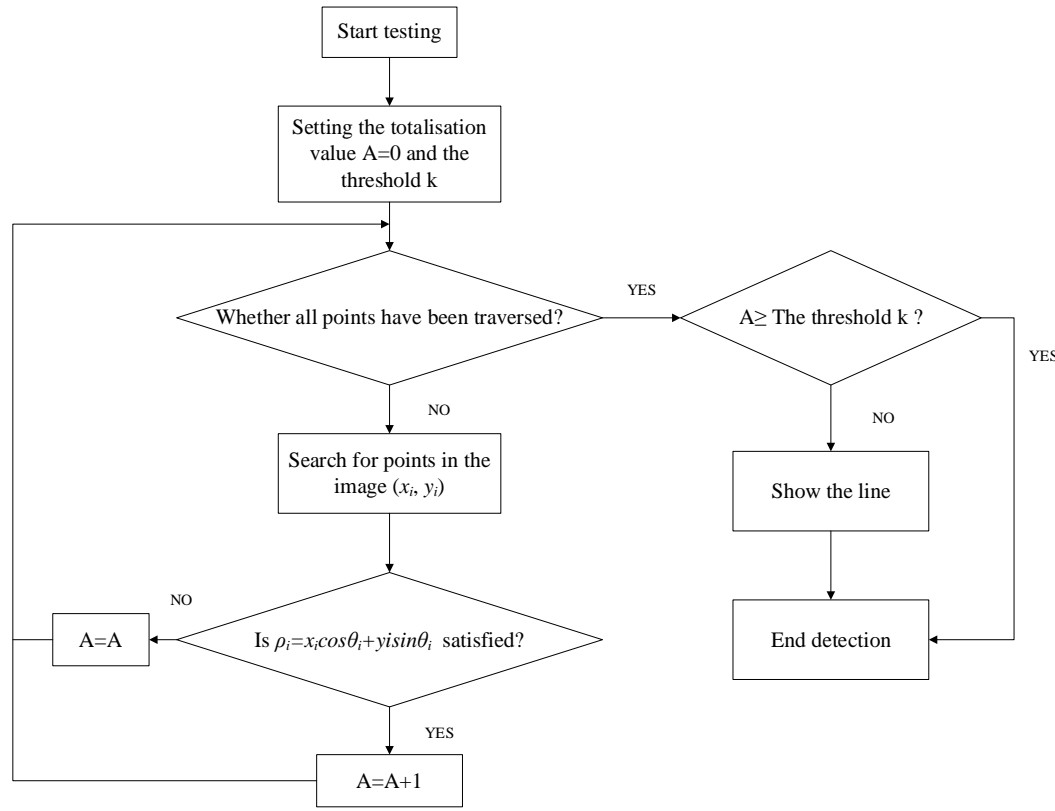

**Figure 6.** Hough transform linear detection flowchart.

(3)  Conveyor belt edge feature extraction test

Belt conveyor belt deviation fault detection mainly uses the Canny edge detection and Hough transform straight line detection algorithms mentioned above. In order to reduce the influence of redundant straight lines on the detection results, ROI region segmentation is also a particularly important part of the region segmentation, which is mainly retained in the region where the conveyor belt is located.

Part of the process and the results of the belt edge straight line detection for a belt conveyor are shown below.

As can be seen from Figure 7, continuous straight lines can be detected at the edge of the conveyor belt in the extraction result, which is an effective extraction result. A reasonable analysis method is chosen to analyze this feature in order to determine the belt bias fault.

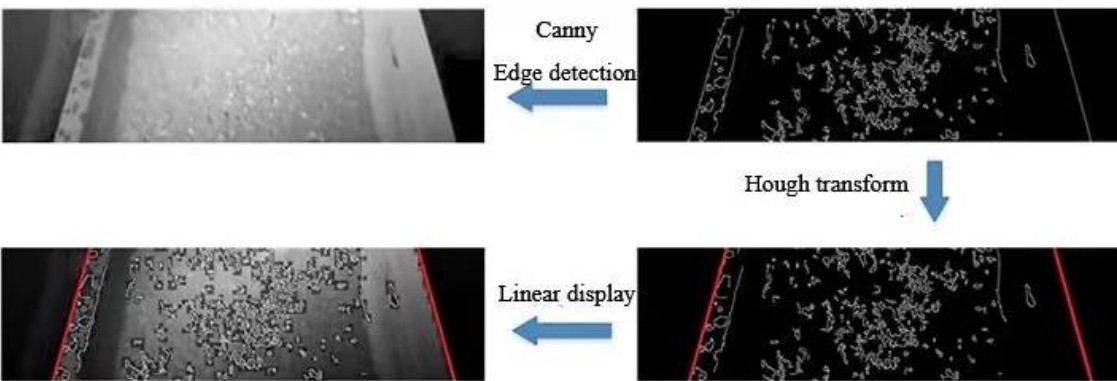

**Figure 7.** Conveyor belt edge feature extraction process.

### 2.3. Feature Identification and Fault Determination

The edge of the conveyor belt straight line with the movement of the conveyor belt is offset and moves, while the belt conveyor frame does not move; that is, the conveyor center baseline position is fixed. This part of the proposed double-baseline belt offset determination method involves analysis of both sides of the conveyor belt relative to the center of the conveyor offset ratio to determine the belt offset fault. When the offset ratio is more than 5%, there is inferred to be occurrence of a belt offset fault. The specific analysis scheme is shown in Figure 8.

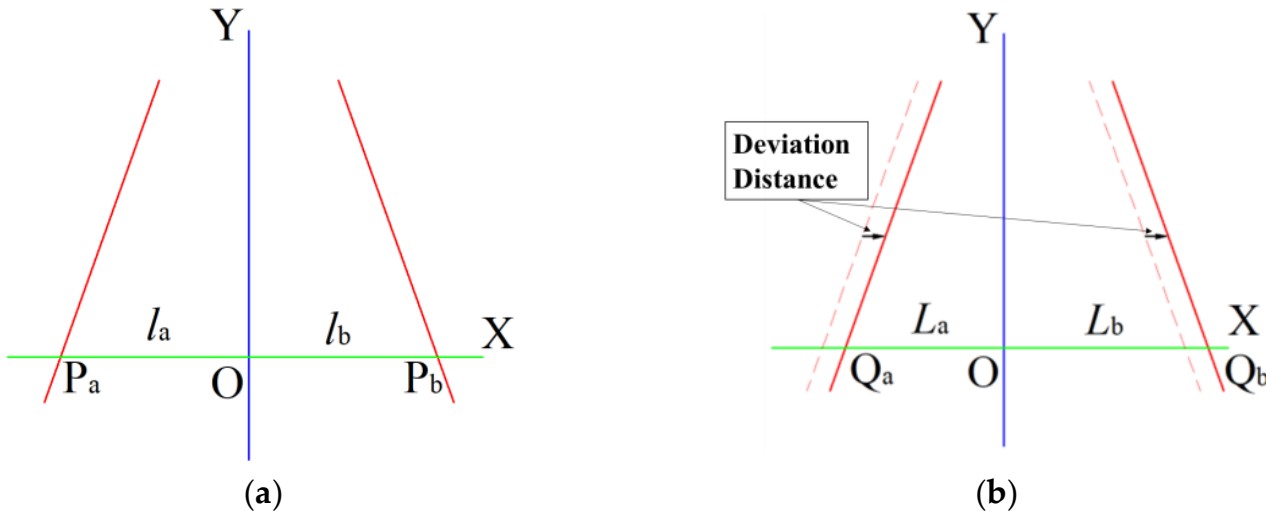

**Figure 8.** Schematic diagram for judgment of band bias fault. (**a**) Normal working condition; (**b**) Faulty condition with deviation.

Figure 8a shows the position of the belt edge under normal working conditions. In the figure, $Y$ is the conveyor center baseline, $X$ is the belt deviation fault detection baseline, point $P_a$ is the intersection of the left edge of the conveyor belt and the detection baseline, $P_b$ is the intersection of the right edge of the conveyor belt and the detection baseline, and $l_a$ and $l_b$ are the widths of the conveyor belt, which are located on both sides of the center baseline $Y$. At this time, it should be satisfied that $l_a = l_b$; that is, the point $P_a$ and the point $P_b$ are symmetric about the center baseline. Figure 8b shows the belt edge when belt deflection occurs. Moving from the dotted line to the solid line, the points $Q_a$ and $Q_b$ are the positions of $P_a$ and $P_b$ after moving, and $L_a$ and $L_b$ are the corresponding belt widths on both sides. The total width of the belt remains unchanged, so only the sizes of $L_a$ and $L_b$ need to be analyzed in order to obtain the deflection proportion of the conveyor belt.

Based on the above analysis, the band bias fault judgment can be carried out using Equations (10) and (11):

$$Lengh = L_a + L_b \tag{10}$$

$$ratio = \begin{cases} (L_b - Lengh/2)/Lengh, L_a < L_b \\ (L_a - Lengh/2)/Lengh, L_a > L_b \end{cases} \tag{11}$$

In the formula, *Lengh* is the width of the conveyor belt located at the baseline unit pixel, and *ratio* is the ratio of the conveyor belt offset and the width of the conveyor belt. When the *ratio* is greater than 5%, it is inferred that a belt conveyor belt deviation fault has occurred.

The results after the detection of the straight line shown in Figure 8 are then analyzed and processed as described above, with the baseline plotted and intersected as shown in Figure 9.

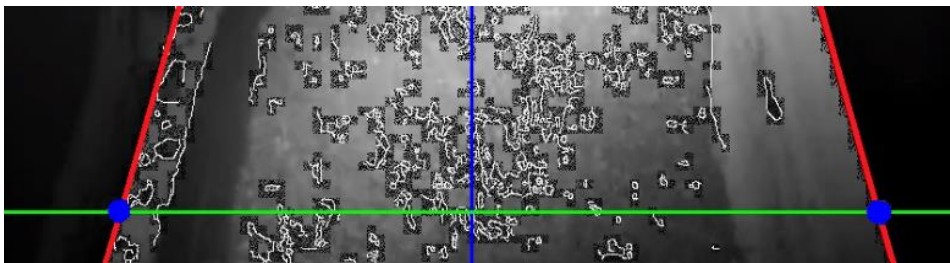

**Figure 9.** Distance detection on both sides of the conveyor belt.

In Figure 9, the blue line is the conveyor center baseline, the green line is the conveyor belt deviation fault detection baseline, and the red line is the belt boundary line. The image resolution is $835 \times 225$. According to the detection results, the intersection point of both sides of the conveyor belt and the baseline is located at the pixel points (112,180) and (780,180). The width of the conveyor belt at the baseline is calculated to be *Lengh* = 620 pixel, and the bandwidth of both sides is La = 300 pixel. The bandwidth of the two sides is $L_b$ = 368 pixel, which can be obtained by the ratio of the belt deviation *ratio* = 5.4%. A ratio of belt deviation greater than 5% is inferred to indicate the occurrence of a belt deviation fault, and at the same time, the validity of the method to determine the belt deviation fault is verified.

## 3. Experimental Design

### 3.1. Experimental Program Design

This study is an experimental study of belt bias fault detection for underground belt conveyors in coal mines. The experiment uses a 720p ordinary industrial camera, which has a low cost and high compatibility. The development tool chosen for this study is Visual Studio 2017 with the OpenCV 4.2.0 open-source computer vision library and the Intel RealSense SDK 2.0 development toolkit.

Coal mine underground lighting conditions are poor. In order to ensure the quality of the acquired images, auxiliary lighting of the shooting environment is required. Different lighting methods affect the quality of the acquired images. Inappropriate angle and intensity of the light source may result in the presence of shadows or reflections in the image, which introduces a lot of noise to the image, reducing the contrast between the captured object and the background, and reducing the quality of the image, while increasing the difficulty of image analysis.

Based on the detection conditions, the scheme shown in Figure 10 is designed, where an industrial camera is mounted directly above the coal flow through a gantry bracket. The photography occurs towards the conveyor belt to ensure the monitoring range. The auxiliary light source uses explosion-proof lighting, with direct lighting to provide auxiliary lighting to the front of the belt conveyor in the coal mine. The environment is poor in the area close to the comprehensive mining face, so it is possible to choose the transshipment place or the place near the coal bunker where faults are likely to occur as the installation location. Multiple detection points can be set up in relation to the detection demand. The above cameras and the light sources are used for the simulation and analysis in this study. The actual application needs to meet the coal mine safety standards.

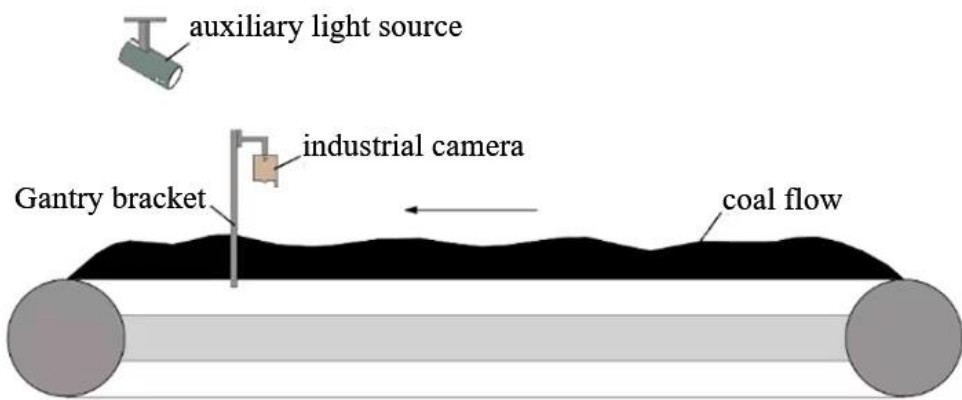

**Figure 10.** Bandwidth detection camera and lighting scheme.

*3.2. Experimental Bench Construction*

In order to verify the effectiveness of the method proposed in this study for belt deflection detection of underground belt conveyors in coal mines, testing simulation experiments were carried out. Consistent with the conditions required for experimental testing, a small conveyor was designed as the experimental platform, as shown in Figure 11.

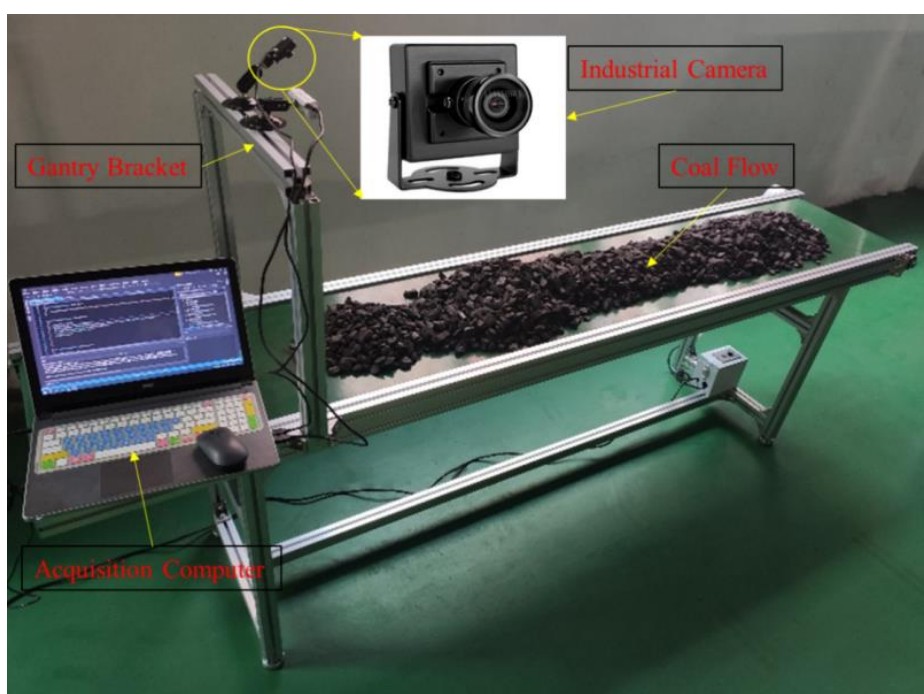

**Figure 11.** Detection experiment platform.

Considering the economy of the experiment, the camera selected was an ordinary 720p industrial camera, which was mounted directly above the belt using a gantry. The maximum resolution of the camera was 1280 × 720, and the frame rate of the captured image was 30 fps.

This part of the experiment occurs on the ground. The shooting environment is open and the lighting conditions are good, so the default image quality of the downhole image can be approximated to that of the experimental image after the processing in the previous steps. The poor conditions of the downhole are not yet set. This part of the experiment is directly based on the captured images to enable the subsequent steps of image information extraction and analysis, as well as testing of the fault detection effect.

The images required for the belt bias fault experiment were taken with the camera facing the conveyor belt area. As shown in Figure 12, the images of the conveyor belt were taken in two states: when no belt bias fault occurred in the experimental setup and when a belt bias fault occurred.

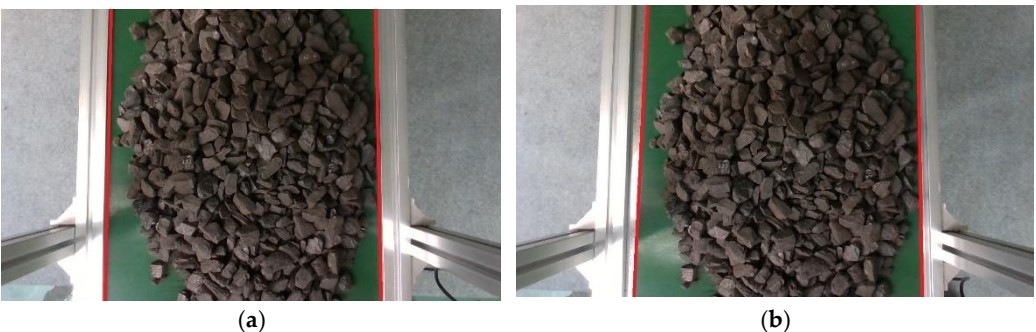

| | |
|:---:|:---:|
| (**a**) | (**b**) |

**Figure 12.** Different states of the conveyor belt. (**a**) No deflection faults; (**b**) Deflection faults occurring.

In order to verify the effectiveness of the research method and get closer to the actual working condition of the conveyor, this part of the experiment was carried out based on the conveyor belt loaded with coal. The coal used for the test was 1/3 coking coal. The detection effect of the edge features of the conveyor belt in this state is shown in Figure 13.

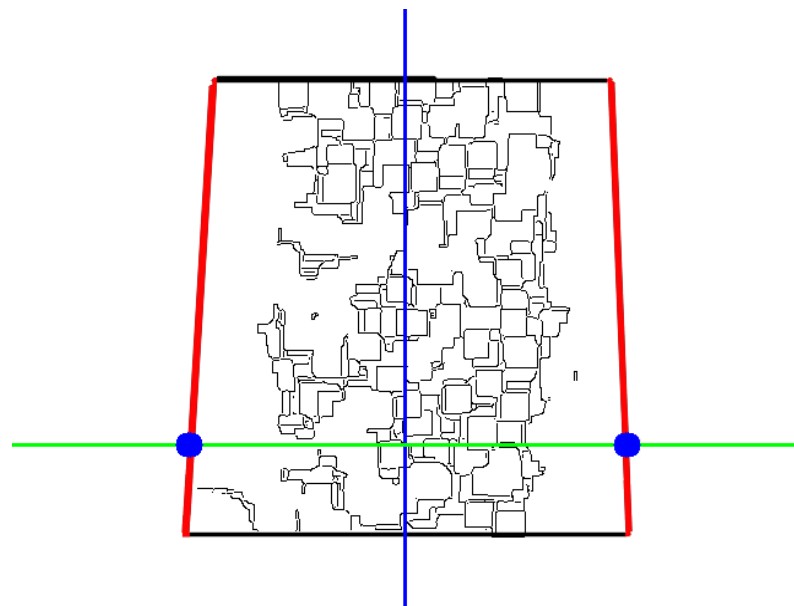

**Figure 13.** Effect of conveyor belt edge detection in the loaded state.

## 4. Results and Discussion

In order to effectively test the effectiveness of this study on belt offset fault detection, experiments were conducted in two states: with the offset ratio within 5% and between 5% and 10%, respectively. For the images under the different offsets, the offset size of the conveyor belt was first measured manually. Then the offset ratio is detected by the system. The difference between the detected offset ratio and the actual offset ratio is compared. The test results are shown in Table 1. First, the relative deviation error is obtained by dividing the sum of the detection error by the sum of the actual bias ratio. Then subtract the relative offset error from 1 to get the detection accuracy.

**Table 1.** Conveyor belt position detection results when the offset ratio is within 5 percent.

| Experiment Number | 1 | 2 | 3 | 4 | 5 | 6 | 7 | 8 |
|---|---|---|---|---|---|---|---|---|
| Actual bandwidth/mm | 700 | 700 | 700 | 700 | 700 | 700 | 700 | 700 |
| Actual deviation distance/mm | 0 | 5 | 10 | 15 | 20 | 25 | 30 | 35 |
| Actual bias ratio | 0 | 0.71% | 1.43% | 2.14% | 2.86% | 3.57% | 4.29% | 5.00% |
| The number of bandwidth pixels | 468 | 468 | 470 | 469 | 470 | 468 | 471 | 470 |
| The number of deviation pixels | 2 | 5 | 8 | 11 | 14 | 17 | 21 | 24 |
| Detecting bandwidth bias ratio | 0.43% | 1.07% | 1.70% | 2.35% | 2.98% | 3.63% | 4.46% | 5.11% |
| Detection error | 0.43% | 0.36% | 0.27% | 0.21% | 0.12% | 0.06% | 0.17% | 0.11% |

Based on the results of Table 1, a graphical comparison of the actual bandwidth and detected bandwidth ratios was performed, as shown in Figure 14.

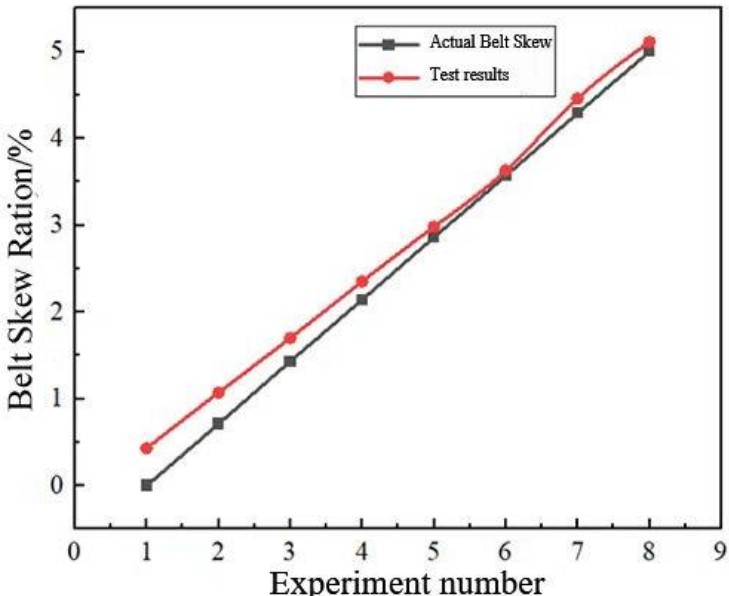

**Figure 14.** Comparison of actual and detected ratios within 5% of the offset ratio.

As can be seen from Table 1 and Figure 14, the actual measured belt deflection ratio gradually increased. The belt deflection ratio detected by the system also showed an increasing trend. The detection results were slightly larger than the actual value, with a maximum absolute error of 0.43%. The errors were all less than 1%, and they decreased with increase in the belt deflection ratio. Based on the above results, experimental detection was carried out for conveyor belts with offset ratios between 5% and 10%. The comparison is shown in Table 2.

**Table 2.** Conveyor belt position detection results when the offset ratio is 5% to 10%.

| Experiment Number | 1 | 2 | 3 | 4 | 5 | 6 | 7 | 8 |
|---|---|---|---|---|---|---|---|---|
| Actual bandwidth/mm | 700 | 700 | 700 | 700 | 700 | 700 | 700 | 700 |
| Actual deviation distance/mm | 40 | 45 | 50 | 55 | 60 | 65 | 70 | 75 |
| Actual bias ratio | 5.71% | 6.43% | 7.14% | 7.86% | 8.57% | 9.29% | 10.00% | 10.71% |
| The number of bandwidth pixels | 471 | 470 | 469 | 470 | 468 | 470 | 471 | 470 |
| The number of deviation pixels | 27 | 31 | 33 | 37 | 40 | 42 | 43 | 45 |
| Detecting bandwidth bias ratio | 5.73% | 6.38% | 7.04% | 7.87% | 8.55% | 9.36% | 9.98% | 10.64% |
| Detection error | 0.02% | 0.05% | 0.10% | 0.01% | 0.02% | 0.07% | 0.02% | 0.07% |

Based on the results of Table 2, a graphical comparison of the actual bandwidth and detected bandwidth ratios was performed, as shown in Figure 15.

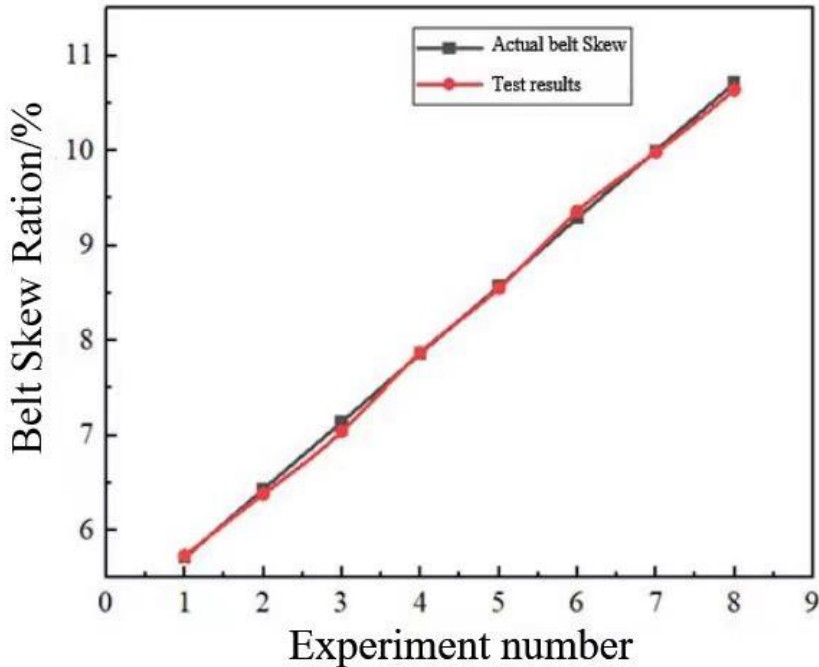

**Figure 15.** Comparison of actual and detected ratios for offset ratios of 5 to 10 percent.

As can be seen from Table 2 and Figure 15, the detected belt deviation results were still in line with the above law. The error was not increased with increase in the ratio and fluctuated above and below the actual value with increase in the ratio of the belt deviation. When the belt deviation ratio is within 5%, the detection accuracy can reach 91.35%, and when the deviation t ratio is between 5% and 10%, the detection accuracy can reach 99.45%. In Wang's paper on belt conveyor deviation detection based on machine vision, the detection accuracy reached 99.6% [18]. One possible reason is that his images had a higher resolution of $1920 \times 1080$. In this experiment, the conveyor belt and the frame of the small conveyor used for the test were considered not to be in the same plane. So, when the offset state was artificially created, the conveyor belt bent, and the detection value of the total width of the conveyor belt decreased. At the same time, because of the light in the experimental environment, the image deviated from the actual edges when straight line fitting was carried out, which, therefore, caused the detection results to be inaccurate. At the time when the artificially created extent of belt deviation was lower, the detection of the ratio of the fluctuation value of the offset to the actual value was larger, so the detection error was larger when the offset was lower.

## 5. Conclusions

Aiming to address the problems that exist in the traditional belt deflection detection technology of underground coal mine belt conveyors, this paper proposes a machine-vision-based belt deviation detection method for underground coal mine belt conveyors. Firstly, a global adaptive high-dynamic-range imaging method is used to perform brightness enhancement processing on the collected photos of the conveyor. Then the straight-line features of the conveyor belt edges are extracted by Canny edge detection and using the Hough transform algorithm. Finally, a dual-baseline positioning judgment method is proposed, which is used to achieve the identification of conveyor belt runout faults. Detection experiments were carried out for different offsets, and the accuracy of belt deflection detection reached 99.45%. A stable and efficient belt conveyor belt offset detection method is obtained in this study. It improves the reliability of underground belt conveyor

monitoring systems and effectively avoids the problem of fault leakage due to the failure of traditional detection devices. However, the detection error in this study is relatively large at low offsets. In a follow-up study, it is intended to address the problem and to propose an effective method of correcting the deviation to ensure the stable and safe operation of belt conveyors.

**Author Contributions:** Conceptualization, X.W. and C.W.; methodology, Z.T.; software, X.H.; validation, X.H., Z.T. and Q.W. data curation, X.W.; writing—original draft preparation, X.W.; writing—review and editing, Z.T.; project administration, X.W. All authors have read and agreed to the published version of the manuscript.

**Funding:** This research was supported by the National Natural Science Foundation of China (52005426, 52375069, 52305078), Major Basic Research Project of the Natural Science Foundation of the Jiangsu Higher Education Institutions (22KJA460013).

**Data Availability Statement:** The datasets generated during, and/or analyzed during, the current study are not publicly available due to privacy requirements. but are available from the corresponding author on reasonable request.

**Conflicts of Interest:** The authors declare no potential conflict of interest with respect to the research, authorship, and/or publication of this article.

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
