# Peer review of "Research on Belt Deviation Fault Detection Technology of Belt Conveyors Based on Machine Vision"

_machines, doi:10.3390/machines11121039_

Round 1
Reviewer 1 Report
Comments and Suggestions for Authors
The references cited in the article have problems, please look and change, there are several sentences with the format: "Error! Reference source not found"
All variables should be described before/after the equation
The authors have to add the state-of-the art references in the manuscripts like us:
Wavelet group method of data handling for fault prediction in electrical power insulators
The perceptron algorithm with uneven margins based transfer learning for turbofan engine fault detection
Modified performance-enhanced PCA for incipient fault detection of dynamic industrial processes
Could the authors explain the key steps involved in the proposed method, including the use of global adaptive high-dynamic-range imaging, edge detection, and dual baseline positioning judgment?
What was the accuracy of the belt deflection detection achieved in the detection experiments mentioned in the conclusion, and why is this accuracy significant?
Change Summary -> Conclusion
How does the proposed method improve the reliability of the underground belt conveyor monitoring system, and what issue does it help to avoid?
The conclusion mentions that there is a limitation in the study related to detection errors at low offsets. What is the significance of this limitation, and what is the plan for addressing it in future research?
Comments on the Quality of English Language
Article with good quality in English, except for the references that were not linked appropriately within the manuscript. Please review.
Author Response
Dear Editors and Reviewers:
Thanks for your letter and reviewers’ comments concerning our manuscript entitled “Research on Belt Deviation Fault Detection Technology of Belt Conveyor Based on Machine Vision” (ID: machines-2720232). These comments are all valuable and very helpful for revising and improving our paper, as well as the important guiding significance to our research. We tried our best to improve the manuscript and made some changes in the manuscript.
The main corrections in this paper and responses to the reviewers’ comments are as follows:
Reviewer #1:
- The references cited in the article have problems, please look and change, there are several sentences with the format: "Error! Reference source not found"
Response: Thank you for your suggestion. We have made changes in the original manuscript. Thank you!
- All variables should be described before/after the equation
Response: Thank you for your suggestion. We have described all variables after the equation in the original manuscript. Thank you!
- The authors have to add the state-of-the art references in the manuscripts like us:
Response: Thank you for your suggestion. We have add the state-of-the art references in the original manuscripts. Thank you!
- Could the authors explain the key steps involved in the proposed method, including the use of global adaptive high-dynamic-range imaging, edge detection, and dual baseline positioning judgment?
Response: Thank you for your positive comments on our research. Global adaptive high-dynamic-range imaging is to realize the brightening process of the image by restoring the details of the dark part of the image. Edge detection is carried out by the following four steps: (1) Gaussian filtering of the image, (2) calculating the gradient value and direction, (3) non-maximum suppression, (4) double threshold filtering and edge connection. Through the above steps to achieve accurate image edge acquisition. On the basis of edge detection, double base line positioning judgment is an algorithm to detect and display straight lines in the graph. The basic principle is to convert the point on the edge line from the rectangular coordinate system to the polar coordinate system, and convert the line in the edge line to a point in the polar coordinate system. The schematic diagram is shown as Fig. 5 in the original manuscripts. Thank you!
- What was the accuracy of the belt deflection detection achieved in the detection experiments mentioned in the conclusion, and why is this accuracy significant?
Response: Thank you for your positive comments on our research. The accuracy of belt offset detection is 1 pixel, as shown in Table 1. This size determines whether the belt machine deviation can be detected. If the value is too large, the belt has actually gone off course, but the camera cannot detect it. So this precision is important. Thank you!
- Change Summary -> Conclusion
Response: Thank you for your suggestion. We have made changes in the original manuscript. Thank you!
- How does the proposed method improve the reliability of the underground belt conveyor monitoring system, and what issue does it help to avoid?
Response: Thank you for your positive comments on our research. The method can detect whether the underground belt conveyor has deviation phenomenon through the monitoring system. Based on the above detection results, the anti-deviation device can be controlled to restore the deviation state. Belt machine deviation will reduce the service life of the belt, easy to lead to belt rupture, increase the failure rate of equipment, but also easy to cause coal dumping. Thank you!
- The conclusion mentions that there is a limitation in the study related to detection errors at low offsets. What is the significance of this limitation, and what is the plan for addressing it in future research?
Response: Thank you for your positive comments on our research. When the deviation of the belt conveyor is low, limited by the pixel of the camera, the mine lighting environment and the installation of the belt conveyor and other influencing factors, small interference will lead to a large detection error. In the future, we plan to use a higher pixel camera and increase the brightness of the lights. In a more ideal environment to observe whether can achieve lower detection error under the condition of low offset. Thank you!
Thank you very much for your comments and suggestions. And here we did not list the changes but marked in yellow in the revised manuscript. We appreciate for editors’ and reviewers’ warm work earnestly and hope that the correction will meet with your approval.
With best regards,
Zuzhi Tian
School of Mechatronic Engineering, China University of Mining and Technology
No.1 Daxue Road, Xuzhou, Jiangsu Province, 221116, P.R. China
E-mail: tianzuzhi2023@163.com

Reviewer 2 Report
Comments and Suggestions for Authors
1. What are the key problems associated with traditional belt deflection detection devices used in underground belt conveyors in coal mines?
2. What is the main objective and novelty of the proposed belt deviation detection method based on machine vision?
3. How does the method utilize global adaptive high dynamic range imaging to enhance the brightness of underground images, and why is this step important?
4. Which algorithms are employed to extract straight-line features of the conveyor belt edges in the machine vision-based method, and what is their role in the detection process?
5. The obtained results can be compared with other algorithms to validate the performance.
6. What is the dual-baseline localization judgment method, and how does it contribute to identifying belt bias faults?
7. What steps were taken to validate the accuracy of the proposed belt deviation detection method, and what level of accuracy was achieved in testing experiments?
8. How does the introduction of this method improve the reliability of belt deviation fault detection in underground belt conveyors in coal mines?
9. Literature support is needed for the discussion part.
10. Kindly add the conclusion.
Comments on the Quality of English LanguageModerate editing of the English language required
Author Response
Dear Editors and Reviewers:
Thanks for your letter and reviewers’ comments concerning our manuscript entitled “Research on Belt Deviation Fault Detection Technology of Belt Conveyor Based on Machine Vision” (ID: machines-2720232). These comments are all valuable and very helpful for revising and improving our paper, as well as the important guiding significance to our research. We tried our best to improve the manuscript and made some changes in the manuscript.
The main corrections in this paper and responses to the reviewers’ comments are as follows:
Reviewer #2:
1.What are the key problems associated with traditional belt deflection detection devices used in underground belt conveyors in coal mines?
Response: Thank you for your positive comments on our research. Most of the traditional belt conveyor belt deviation fault detection devices use mechanical triggered structural changes to complete the detection work, only when the belt deviation to a certain degree can be detected. This method can not be detected at low deviation, which will still reduce the service life of the belt, and there is a problem of coal sprinkling. And the traditional belt machine deviation device is easy to fail. The reliability is low. Thank you!
2. What is the main objective and novelty of the proposed belt deviation detection method based on machine vision?
Response: Thank you for your positive comments on our research. This study is used to solve the problem that the traditional belt conveyor belt deviation fault detection device is easy to fail. The detection range is larger, and it can be detected at low deviation degree. Moreover, the detection method based on the camera is easier to be found when the camera fails, which reduces the cost of detecting whether the sensor fails from another aspect. Thank you!
3. How does the method utilize global adaptive high dynamic range imaging to enhance the brightness of underground images, and why is this step important?
Response: Thank you for your positive comments on our research. This method brightens the images of underground belt conveyor and coal mine roadway, and restores the details of the dark part. When the brightness of each point of the picture is known, the maximum brightness and the average value are calculated. The calculation formula is shown in formula 1. Because the underground environment of the mine is very dark, the photos taken are relatively dark. If the brightness is not enhanced, it may lead to an increase in the error of the edge detection of the belt machine, and even cannot identify the edge of the belt machine. Thank you!
4. Which algorithms are employed to extract straight-line features of the conveyor belt edges in the machine vision-based method, and what is their role in the detection process?
Response: Thank you for your positive comments on our research. Canny edge detection and Hough transform line detection are used to extract the straight-line features of conveyor belt edges. Canny edge detection is used to extract the features of both sides of the conveyor belt edge. Hough transform line detection is to detect and display straight lines in the graph on the basis of edge detection. Thank you!
5. The obtained results can be compared with other algorithms to validate the performance.
Response: Thank you for your suggestion. We have added a comparison with other papers in the discussion section. Thank you!
6. What is the dual-baseline localization judgment method, and how does it contribute to identifying belt bias faults?
Response: Thank you for your positive comments on our research. The method first calculates the distance between the two sides of the belt and the central base line of the conveyor. When the belt deviation occurs, the deviation rate is calculated based on the larger distance. The specific calculation formulas are shown in (11) and (12). The schematic diagram is shown in Fig. 8. The dual-baseline localization judgment method is a method of calculating the offset ratio of the belt, converting the image to the offset ratio. When the offset ratio exceeds 5%, the biased fault is judged to occur, which is conducive to identifying the biased fault. Thank you!
7. What steps were taken to validate the accuracy of the proposed belt deviation detection method, and what level of accuracy was achieved in testing experiments?
Response: Thank you for your positive comments on our research. Firstly, the actual deviation distance is measured, and the actual band deviation ratio is calculated according to the distance. Then, according to the visual method proposed in this paper, the detection band offset ratio is obtained. The detection error is obtained by comparing with the actual band offset ratio. The method is directly compared with the actual offset ratio. And the accuracy of the detection error obtained is very high, which can directly reflect the detection effect of the method. Thank you!
8. How does the introduction of this method improve the reliability of belt deviation fault detection in underground belt conveyors in coal mines?
Response: Thank you for your positive comments on our research. The method can detect whether the underground belt conveyor has deviation phenomenon through the monitoring system. The control center can control the anti-deviation device to restore the deviation state based on the above detection results. The traditional belt conveyor deviation control device is easy to fail and has low reliability. And when the traditional device fails, it is difficult to find its failure without a safety check. However, this problem can be avoided by monitoring methods, and once the monitoring fails, it can be known by the display screen. Thank you!
9. Literature support is needed for the discussion part.
Response: Thank you for your suggestion. We have added literature support in the discussion section. Thank you!
10. Kindly add the conclusion.
Response: Thank you for your suggestion. We have added the conclusion. Thank you!
Thank you very much for your comments and suggestions. And here we did not list the changes but marked in yellow in the revised manuscript. We appreciate for editors’ and reviewers’ warm work earnestly and hope that the correction will meet with your approval.
With best regards,
Zuzhi Tian
School of Mechatronic Engineering, China University of Mining and Technology
No.1 Daxue Road, Xuzhou, Jiangsu Province, 221116, P.R. China
E-mail: tianzuzhi2023@163.com
Reviewer 3 Report
Comments and Suggestions for Authors
This paper proposes a belt deviation detection method based on machine vision. In review, there are several comments & concerns as below:
1. The state-of-the-art is marginal and the authors must work on extending this section. Motivation and novelty should be clearly emphasized. Currently, it is below average in the current manuscript.
2. Most of the places do not show the cited reference. May be an error!
3. None of the used equations have been cited.
4. What kind of calibration needs to be done for accurate measurements and prediction?
5. Does the speed of the belt and loading influence the detection results?
6. What about the effect of creep in the belt? The authors did not comment anything about it.
5. The main important concern is the vibration. However, there is no mention of it and how its influence can be mitigated.
6. What if the loading is scattered and the edge for a certain portion is not visible for the camera to capture?
7. The quality of figures and overall presentation mus be improved.
Author Response
Dear Editors and Reviewers:
Thanks for your letter and reviewers’ comments concerning our manuscript entitled “Research on Belt Deviation Fault Detection Technology of Belt Conveyor Based on Machine Vision” (ID: machines-2720232). These comments are all valuable and very helpful for revising and improving our paper, as well as the important guiding significance to our research. We tried our best to improve the manuscript and made some changes in the manuscript.
The main corrections in this paper and responses to the reviewers’ comments are as follows:
Reviewer #3:
This paper proposes a belt deviation detection method based on machine vision. In review, there are several comments & concerns as below:
1.The state-of-the-art is marginal and the authors must work on extending this section. Motivation and novelty should be clearly emphasized. Currently, it is below average in the current manuscript.
Response: Thank you for your positive comments on our research. We really don't have any great discoveries in this paper. And the introduction part of the original manuscript did not write the novelty and importance clearly. In light of this, we have strengthened the introduction to highlight innovation. Thank you!
2.Most of the places do not show the cited reference. May be an error!
Response: Thank you for your suggestion. We have changed the references cited in the original manuscript. Thank you!
3. None of the used equations have been cited.
Response: Thank you for your suggestion. We have added a reference to the equation. Thank you!
4. What kind of calibration needs to be done for accurate measurements and prediction?
Response: Thank you for your positive comments on our research. In order to obtain accurate measurements, we first ensured the vertical of the gantry bracket and belt holder when installing the entire test bench. This can ensure the stability of industrial camera installation. And multiple photos will not produce position changes. When obtaining the actual offset distance, we measure the distance at the same horizontal line to ensure the validity of the data. Thank you!
5. Does the speed of the belt and loading influence the detection results?
Response: Thank you for your positive comments on our research. The speed and load of the belt may affect the deviation distance of the belt. However, in the actual process, the belt conveyor is basically in a stable state, so the deviation distance of taking photos is basically stable. In this study, we first set the actual deviation distance, and then detect the camera system to obtain the offset ratio respectively. This is basically in line with the actual scenario. Thank you!
6. What about the effect of creep in the belt? The authors did not comment anything about it.
Response: Thank you for your positive comments on our research. The creep of the belt will deform the belt, so that the edge of the belt is not straight. This is really not considered in the paper. However, after the belt is creeped, there is a safety hazard when the belt is used again, so the creep belt needs to be replaced. It can then be studied how machine vision identifies the creep of the belt. Thank you!
7. The main important concern is the vibration. However, there is no mention of it and how its influence can be mitigated.
Response: Thank you for your positive comments on our research. For belt conveyors, vibration is indeed a serious problem. However, in a single camera picture, the picture is still, and the bias ratio is determined. Therefore, it is possible to detect the band detection ratio by machine vision method in this paper. The subsequent research can consider the influence of vibration on the belt deviation of belt conveyor. Thank you!
8. What if the loading is scattered and the edge for a certain portion is not visible for the camera to capture?
Response: Thank you for your positive comments on our research. That the loading is scattered is a possible situation. But the belt conveyor is moving, and the deviation is basically unchanged. The coal does not always press down on the edge of the belt, so it can be detected elsewhere. Therefore, when the coal presses the edge more, the subsequent research can be made on this situation, and the alarm can be made in this case. Thank you!
9. The quality of figures and overall presentation must be improved.
Response: Thank you for your positive comments on our research. We have made changes in the original text in terms of language and data. Thank you!
Thank you very much for your comments and suggestions. And here we did not list the changes but marked in yellow in the revised manuscript. We appreciate for editors’ and reviewers’ warm work earnestly and hope that the correction will meet with your approval.
With best regards,
Zuzhi Tian
School of Mechatronic Engineering, China University of Mining and Technology
No.1 Daxue Road, Xuzhou, Jiangsu Province, 221116, P.R. China
E-mail: tianzuzhi2023@163.com
Round 2
Reviewer 2 Report
Comments and Suggestions for Authors
1 Line 153, kindly cite the reference properly.
Congrats to the authors.
Comments on the Quality of English LanguageMinor editing of the English language required
Reviewer 3 Report
Comments and Suggestions for Authors
The manuscript can be accepted in its present form.